# Pre-existing antibody levels negatively correlate with antibody titers after a single dose of BBV152 vaccination

Suman Das[1,6], Janmejay Singh[2,6], Heena Shaman[2,6], Balwant Singh[2], Anbalagan Anantharaj[2], Patil Sharanabasava[2,5], Rajesh Pandey[3], Rakesh Lodha[4], Anil Kumar Pandey[1] & Guruprasad R. Medigeshi [2✉]

Many adults in India have received at least one dose of COVID-19 vaccine with or without a prior history SARS-CoV-2 infection. However, there is limited information on the effect of prior immunity on antibody response upon vaccination in India. As immunization of individuals continues, we aimed to assess whether pre-existing antibodies are further boosted by a single dose of BBV152, an inactivated SARS-CoV-2 vaccine, and, if these antibodies can neutralize SARS-CoV-2 Delta and Omicron variants. Here we show that natural infection during the second wave in 2021 led to generation of neutralizing antibodies against other lineages of SARS-CoV-2 including the Omicron variant, albeit at a significantly lower level for the latter. A single dose of BBV152 boosted antibody titers against the Delta and the Omicron variants but the antibody levels remained low against the Omicron variant. Boosting of antibodies showed negative correlation with baseline neutralizing antibody titers.

[1] Employees State Insurance Corporation Medical College and Hospital, Faridabad, Haryana, India. [2] Translational Health Science and Technology Institute, Faridabad, Haryana, India. [3] INtegrative GENomics of HOst-PathogEn (INGEN-HOPE) Laboratory, CSIR-Institute of Genomics and Integrative Biology, Delhi, India. [4] All India Institute of Medical Sciences, New Delhi, India. [5] Present address: Serum Institute of India, Pune, India. [6] These authors contributed equally: Suman Das, Janmejay Singh, Heena Shaman. ✉email: gmedigeshi@thsti.res.in

Inactivated vaccines have catered to almost half of the world's COVID-19 vaccine needs[1]. In India, the viral-vectored ChAdOx1 nCoV-19 and the inactivated BBV152 vaccines were approved for use in adults in 2020 and 2021. These two vaccines have been the mainstay of COVID-19 vaccination program in India. Less than 10% of the Indian population had received one dose of COVID-19 vaccine at the onset of second wave of COVID-19 in April 2021. Serosurveillance studies have shown that 69% of the Indian population had antibodies for COVID-19 after the second wave of SARS-CoV-2 infections between April 2021 and June 2021[2]. A subsequent serosurvey from Delhi showed seropositivity of over 90%[3] suggesting that India was/is a heterogeneous mix of people with immunity to COVID-19 due to vaccination or natural infection or both after the second wave of SARS-CoV-2 cases.

There are a number of reports assessing the effect of prior infection on antibody responses post-vaccination. In the case of mRNA vaccines, antibody response in individuals with prior history of natural SARS-CoV-2 infection was at least one order of magnitude higher compared to seronegative individuals[4,5]. Another study has shown that the neutralizing antibody titers varied in the three groups of individuals who were vaccinated with BNT162b2 mRNA vaccine either 1 to 2 months, 2 to 3 months, or more than 3 months after infection. The group which received the first dose of mRNA vaccine more than 3 months after SARS-CoV-2 infection generated higher levels of neutralizing antibodies compared to the other two groups[6]. Similarly, previously-infected individuals receiving a single dose of ChAdOx1 nCoV-19 generated neutralizing antibodies equivalent to that generated after two doses of BNT162b2 vaccine in naive individuals. These antibodies were capable of neutralizing different variants of concern of SARS-CoV-2[7]. In a cohort of healthcare workers in Brazil, about 48% of participants of the study had prior history of SARS-CoV-2 infection at the time of receiving the first dose of CoronaVac, an inactivated whole SARS-CoV-2 vaccine[8]. Anti-spike antibodies measured by ELISA were about five-fold higher after one dose of CoronaVac in participants with prior COVID-19 infection and no further increase in antibodies was observed after the second dose of the vaccine[8].

BBV152 was one of the two vaccines licensed in India and now in over 20 other countries. A larger proportion of people in India got vaccinated after the second wave in 2021. We aimed to measure the impact of pre-existing humoral immunity on response to subsequent vaccination with one dose of BBV152. As antibody levels wane and the risk of reinfection with the same variant and or a new variant warrants booster vaccination, it is imperative to understand the efficiency of boosting vis-à-vis pre-existing antibody levels. In this work, we enrolled adult subjects from the general population who got vaccinated with BBV152 as part of a routine COVID-19 vaccination program and assessed the effect of pre-existing antibody levels on boosting of antibody titers after receiving a single dose of vaccine. We found that most of the individuals had neutralizing antibodies to the Delta variant at the time of receiving their first dose of BBV152 vaccine. Natural infection with the Delta variant led to generation of neutralizing antibodies against other lineages of SARS-CoV-2 including the Omicron variant in a subset of individuals, although the antibody titers were significantly lower for this new variant of concern as compared to the Delta variant and/or the ancestral virus.

## Results

**Prior antibody titers negatively correlate with antibody boosting.** Among the general public who visited the vaccination center to receive their first dose of COVID-19 vaccine (BBV152), a total of 94 adult subjects (40 females) consented to participate in the study. We collected baseline blood sample before the first dose of vaccination and a follow-up sample 4 weeks later, before the administration of the second dose of BBV152 in the months of June–July 2021. Median age of the subjects was 30.5 yrs (range: 18–67 yrs). We interpreted positivity in the ELISA for SARS-CoV-2 spike receptor-binding domain (RBD) as a measure of seropositivity. Sixty-seven of 94 (71.3%) subjects were positive in a quantitative RBD-ELISA of the baseline sample. Four samples were equivocal. After one dose of vaccination, positivity in RBD-ELISA increased to 84 of 94 samples (89.4%). The geometric mean titers (GMT) of RBD antibodies increased significantly from 109 (95% CI: 76, 156) to 206 (95% CI: 163, 260) in subjects who were seropositive at baseline (Fig. 1a). In the case of participants who had a negative RBD-ELISA at baseline ($n = 23$), the GMT of RBD antibodies increased significantly from below the

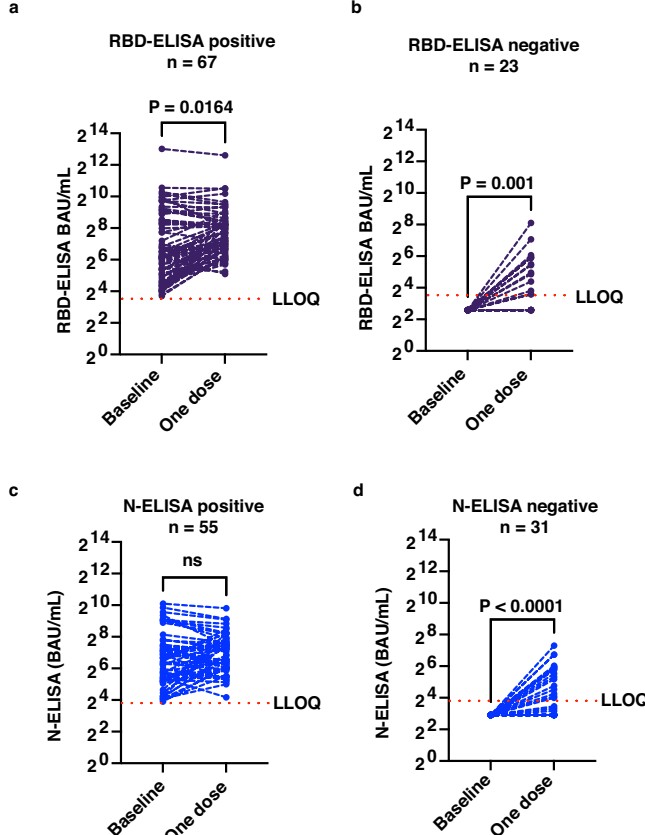

**Fig. 1 Booster effect of a single dose of BBV152 vaccination.** Geometric mean titers of antibodies against SARS-CoV-2 RBD were estimated by quantitative RBD-ELISA and data presented as geometric mean with 95% confidence intervals at baseline and after one dose in **a** RBD-ELISA-positive samples at baseline ($n = 67$ independent values) or **b** RBD-ELISA-negative samples at baseline ($n = 23$ independent values). **c** Geometric mean titers of antibodies against SARS-CoV-2 Nucleocapsid (N) were estimated by quantitative N-ELISA and data presented as GMT at baseline and after one dose in **c** N-ELISA-positive samples at baseline ($n = 55$ independent values) or **d** N-ELISA-negative samples at baseline ($n = 31$ independent values). Data were expressed as binding antibody units (BAU)/ml (international units). Dotted line indicates the lower limit of quantitation (LLOQ) which is 12 BAU/mL for RBD-ELISA and 15 BAU/mL for N-ELISA. Two-tailed P values were estimated by Wilcoxon matched-pairs signed-rank test in **a–d**. ns: Not significant. Source data are provided as a Source data file.

**Table 1 Virus neutralization titers post single dose of BBV152 vaccination based on RBD-ELISA positivity.**

| Sampling time | Geometric mean FRNT50 (95% CI) | | | | | |
|---|---|---|---|---|---|---|
| | Baseline WT (A) | One dose WT (B) | P value comparing the titers within each group; before and after 1 dose of vaccine (A vs B) (Wilcoxon signed-rank test) | Baseline B.1.617.2 (C) | One dose B.1.617.2 (D) | P value comparing the titers within each group; before and after 1 dose of vaccine (C vs D) (Wilcoxon signed-rank test) |
| All (n = 94) | 108.6 (81.3–145.1) | 256.4 (196.1–335.3) | <0.0001 | 291.4 (209.6–405.0) | 508.1 (398.9–647.2) | 0.0126 |
| Groups based on RBD-ELISA results | | | | | | |
| 1. RBD-positive at baseline (n = 67) | 182.8 (133.6–250.0) | 422.2 (340.5–523.5) | 0.0001 | 531.9 (371.6–761.2) | 783.2 (624.8–981.7) | 0.448 |
| 2. RBD-Negative at baseline (n = 23) | 26.50 (19.8–35.5) | 57.32 (33.0–99.6) | 0.0033 | 63.94 (46.4–88.1) | 146.6 (95.7–224.7) | 0.0002 |
| P value comparing the titers between 2 groups based on RBD-ELISA (using Wilcoxon rank-sum test) | <0.0001 | <0.0001 | | <0.0001 | <0.0001 | |

Comparison of GMTs (with 95% CI) of neutralizing antibodies at baseline and 4 weeks after 1st dose was performed using Wilcoxon signed-rank test. Comparison of GMTs between the groups (based on RBD-ELISA results) was performed using the Wilcoxon rank-sum test. Two-tailed P values are indicated.

level of detection at baseline to 19.4 (95% CI: 12, 32) after one dose of vaccination (Fig. 1b). We also measured the levels of nucleocapsid (N) antibodies by quantitative ELISA. Fifty-five of the 94 (58.5%) samples were positive for N antibodies suggesting prior exposure, most likely in the second wave. After one dose of vaccination, positivity in N-ELISA increased to 84% (79 out of 94). Eight samples were equivocal. The increase in the GMT of N antibodies in these 55 seropositive subjects was from 90.7 (95% CI: 66, 124) to 122.5 (95% CI: 98, 153) (Fig. 1c). Seventeen of the 31 baseline N-ELISA-negative subjects became positive (54.8%) after the first dose of vaccination with BBV152. The GMT of N antibodies increased significantly from below the level of detection to 22.5 (95% CI: 15, 34) in these subjects (Fig. 1d).

**Insignificant boosting of neutralizing antibodies to Delta variant.** We next measured the virus-neutralizing antibody titers against the SARS-CoV-2 B.6 lineage virus from 2020[9] and B.1.617.2 (Delta variant) isolates in samples collected at baseline and 4 weeks after first dose of vaccination by focus reduction neutralization titer (FRNT) assay[10]. Overall the GMT of neutralizing antibodies increased after one dose of BBV152 vaccination for both the B.6 virus and the Delta variant (Table 1). Both RBD and N antibody levels significantly correlated with neutralization titers against the Delta variant after the first dose, however, the correlation between the GMT of RBD antibodies and FRNT values was better compared to the same between N antibodies and FRNT values (Supplementary Fig. S1). The FRNT data were further analyzed based on the RBD-ELISA positivity status at baseline. We found that in the 67 RBD-ELISA-positive samples, one dose of BBV152 vaccine led to significant increase in the GMT for neutralizing antibodies against the B.6 virus from 182.8 to 422.2 (Table 1 and Fig. 2a). However, in the same group, the GMT for the Delta variant showed a marginal and insignificant increase from 531.9 to 783.2 (Table 1 and Fig. 2a). Nevertheless, this 1.5-fold increase in GMT and the absolute level of antibodies could still be beneficial in conferring protection in the long run. Seventeen out of 23 baseline RBD-ELISA-negative samples showed the presence of neutralizing antibodies for B.6 and 22 out of 23 samples were positive for neutralizing antibodies

against the Delta variant in the FRNT assay. However, the GMT of neutralizing antibodies in these samples was about seven- to eight-fold lower compared to the ELISA-positive samples (Table 1) suggesting that the antibody levels in these samples were below the level of detection of RBD-ELISA or these are neutralizing antibodies that bind to regions other than the RBD. In the participants with a negative RBD-ELISA at baseline, the GMT of antibodies against both the B.6 and B.1.617.2 lineage viruses increased significantly, from 26.5 to 57.3 for B.6 and from 63.9 to 146.6 for Delta variant, after receiving the first dose of the vaccine (Table 1 and Fig. 2b). The 2.3-fold increase in neutralizing antibody titers for the delta variant in seronegative individuals as compared to the 1.5-fold increase in seropositive individuals suggests that higher pre-existing antibody levels may compromise the boosting of antibodies upon vaccination with BBV152 for the Delta variant. The RBD antigen used in the study was from the ancestral Wuhan strain, therefore, it is plausible that vaccination with BBV152 does lead to increase in binding antibodies to RBD of Wuhan strain which may be inefficient in neutralizing the Delta variant. To test this, we further assessed the ratio of neutralizing antibody titers for the B.6 lineage or the Delta variant to RBD antibody titers in all the samples between baseline and post-one dose vaccination. The median (with interquartile range) FRNT/RBD value in the baseline and post-one dose sample was 2.3 (1.1, 4.7) and 2.4 (1.4, 4.2), respectively (n = 83), for B.6 lineage virus indicating proportionate levels of RBD-binding and neutralizing antibodies for the B.6 lineage virus (Supplementary Fig. S2a). However, in the case of the Delta variant, the median (with interquartile range) FRNT/RBD value was 6.3 (3.3, 13.0) at baseline which reduced significantly to 4.2 (2.7, 8.0) (n = 93) indicating a higher proportion of RBD-binding antibodies as compared to neutralizing antibodies after one dose of vaccination (Supplementary Fig. S2b) which agrees with earlier reports showing reduced neutralization of the Delta variant by many vaccines including BBV152 and reduced vaccine effectiveness[11–16]. Overall, we observed that samples with baseline $FRNT_{50} > 1000$ failed to show an induction in neutralizing antibodies after the first dose. We observed a clear negative correlation between the baseline $FRNT_{50}$ titer for the delta variant

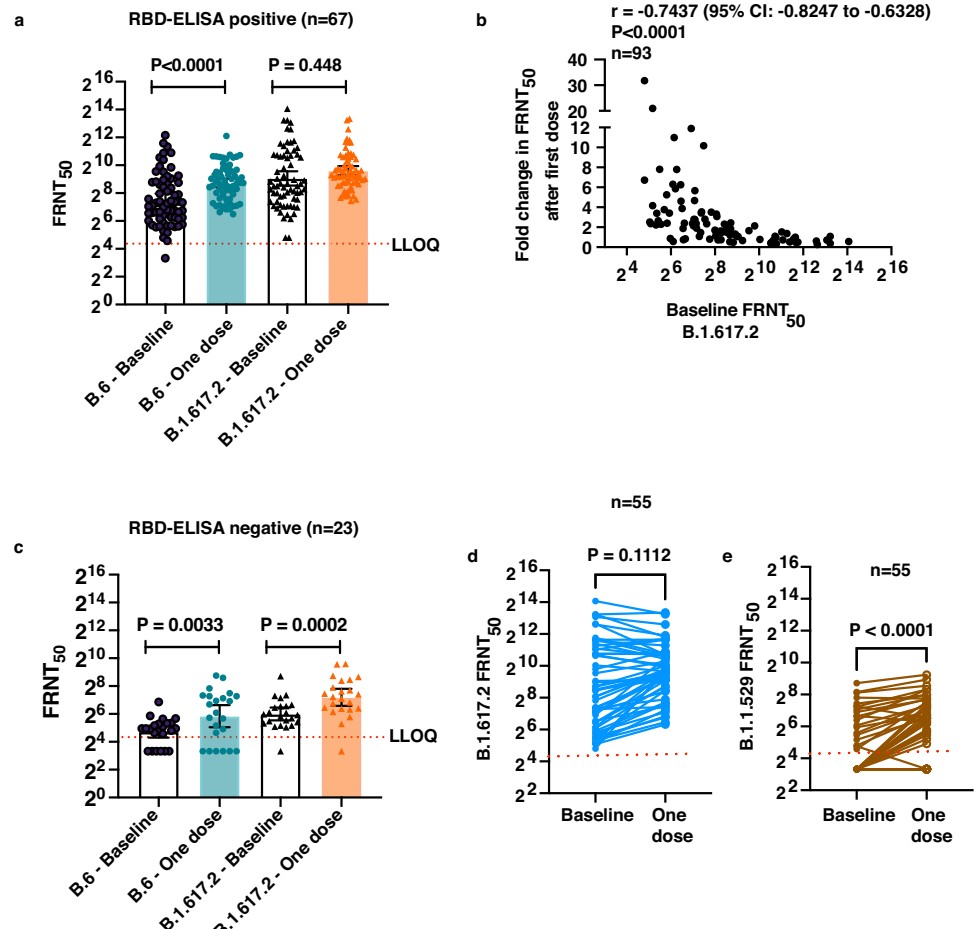

**Fig. 2 Neutralizing antibody titers after single dose of BBV152 vaccination. a** $FRNT_{50}$ titers in RBD-ELISA-positive samples for the ancestral B.6 lineage (circle symbols) and the Delta variant (B1.617.2) (triangle symbols) in baseline (clear bars) and post single- dose (shaded bars) vaccination samples ($n = 67$ independent values). **b** $FRNT_{50}$ titers in RBD-ELISA-negative samples for the ancestral B.6 lineage (circle symbols) and the delta variant (triangle symbols) in baseline (clear bars) and post single-dose (shaded bars) vaccination samples. Data are presented as geometric mean with 95% confidence intervals ($n = 23$ independent values). **c** Spearman correlation (r) between baseline $FRNT_{50}$ titers for B1.617.2 and fold change observed after single dose of BBV152 vaccination ($n = 93$ independent XY pairs) (**d**, **e**) $FRNT_{50}$ titers for the B1.617.2 variant and Omicron (1.1.529) variant in baseline and post single-dose vaccination samples ($n = 55$ independent values). LLOQ: Lower limit of quantitation of the assay. Two-tailed P values were estimated by Wilcoxon matched-pairs signed-rank test. Source data are provided as a Source data file.

and the fold change in titer values after a single dose of vaccination with BBV152 (Fig. 2c).

**Induction of neutralizing antibodies to Omicron variant.** While this study was in progress, Omicron (B.1.1.529) emerged as a variant of concern and antibodies from most vaccines showed reduced efficiency in neutralizing this variant of concern[17–19]. From the 94 samples, we randomly selected 55 paired samples, which had a detectable $FRNT_{50}$ value for the delta variant, to test for their ability to neutralize the Omicron variant. Only twenty of 55 baseline samples had detectable levels of neutralizing antibodies against the Omicron variant. By assigning a $FRNT_{50}$ value of 10 for the samples which had no detectable levels of antibodies in the starting dilution (1:20) of the assay, we obtained a GMT of 22.5 (95% CI: 16, 31) for these 55 samples. This value was 20-fold lower than the GMT of Delta variant which was 404 (95% CI: 248, 658). After a single dose of BBV152, the number of samples positive for neutralizing antibodies against the Omicron variant increased to 36 out of 55 subjects with a significant increase in GMT to 52 (95% CI: 36, 75). However, this value was still

15-fold lower than the GMT for the Delta variant which was 784 (95% CI: 575, 1068) (Fig. 2d, e). Whether the modest boosting in neutralizing antibodies observed for the Omicron variant by one dose of BBV152 vaccination is protective or not warrants further studies from the ongoing booster vaccination campaign across the country.

## Discussion
Our research question was focused on understanding how prior exposure to SARS-CoV-2 affects response to vaccination with BBV152 which mimics the booster dose scenario. The study relied on collecting samples from general subjects who were visiting the vaccination center for routine vaccination. Other studies have shown that vaccination in early convalescent individuals leads to poorer boosting of antibody responses[20–22]. In naturally infected individuals, booster dose after 6 months is predicted to increase the vaccine effectiveness against variants of concern[23]. We were not able to ascertain the exact date of past infection in most of these individuals which is not unusual as most COVID-19 infections are asymptomatic or mild and are not

diagnosed. We detected neutralizing antibodies to both the B.6 lineage (prevalent during early 2020) and to the Delta variant in 22 out of 23 RBD-ELISA-negative samples suggesting past exposure. The overall seroprevalence in India till Feb 2021 was 20.7% which increased to 69.2% by August 2021[2]. Around 11% of the population had received two doses of the vaccine and 25% had received one dose by the end of August 2021. Therefore, the estimated 17 million infections during the second wave would have contributed to increase in seropositivity. Nucleocapsid antibodies are known to decay with a half-life of 68 days[24] and 59% of our study participants were positive for N antibodies suggesting that they could have been infected with SARS-CoV-2 within 6–8 months prior to vaccination. Therefore, we cannot rule out the possibility that some of the subjects with high antibody titers were in their early convalescence and therefore, the boosting effect may not have been significant due to saturation/exhaustion of the immune system as has been observed in other studies[25].

Antibody boosting correlated negatively with the levels of pre-existing antibodies which indicates that a strategy is needed to prioritize high-risk individuals based on their antibody levels for booster vaccination. Nevertheless, the level of neutralizing antibodies for the Omicron variant after boosting in both seropositive and seronegative individuals remained much lower compared to the ancestral virus or the Delta variant which is consistent with recent reports for CoronaVac, an inactivated vaccine[26]. It has been estimated that neutralizing antibodies are a good correlate of protection and contribute to about 60% of the protective efficacy of a vaccine which indicates that the cellular responses play a critical synergistic role along with antibodies in mediating protection from SARS-CoV-2 variants of concern (VoCs)[9,13,23]. Some of the recent studies have also shown that heterologous boosting mounts a robust immune response to VoCs[27]. As many new vaccines are likely to be licensed in the coming months in India, more studies are required to measure the efficacy of homologous vs heterologous boosting against new VoCs.

While this manuscript was under preparation, a study from India reported the antibody responses in individuals with or without prior SARS-CoV-2 infection who received BBV152 vaccine[28]. The ELISA antibody titers (anti-spike IgG) after one dose of BBV152 vaccine in individuals with prior SARS-CoV-2 infection were about two-fold higher as compared to the antibody titers after two doses of the same vaccine in participants with no prior history of infection[28]. The binding antibody titers in ELISA increased by four-fold after one dose of vaccine in individuals with prior SARS-CoV-2 infection which did not show further increase after receiving the second dose. There was no significant increase in neutralization antibody titers for the wild-type virus after one or two doses of BBV152 vaccine in previously-infected individuals[28]. This result provides an independent validation of our observations suggesting that booster vaccination policy should take the antibody levels in individuals into account and prioritize high-risk individuals with low or undetectable levels of antibodies for booster doses.

## Methods
Please see Supplementary Table 1 for the list of key resources and reagents used in this manuscript.

**Human ethics**. The study was approved by the Institutional Ethics Committees For Human Research at ESIC Hospital and Medical College (No. 134/R/10/IEC/22/2021/02) and THSTI (THS 1.8.1/ (93)). Informed consent was obtained from all the participants.

**Human samples**. *BBV152 cohort*: An informed consent was obtained in person from each participant before they were being recruited in this study at the ESIC Medical College & Hospital, Faridabad between May and August 2021. All adults

>18 years of age who came for the first dose of vaccine were eligible to participate in this study including those who had recovered from the COVID-19 in the recent past. In addition, participants were requested to provide any history related to COVID-19 infection. Blood samples of the participants were collected in anti-coagulant-free vacutainers with the help of a professional phlebotomist and stored at 2–8 °C. First blood sample (5 ml) was collected on day 1 prior to the administration of the first dose of the vaccine. Second blood sample (5 ml) was collected during the follow-up visit before the administration of the second dose of the vaccine (4 weeks after the first dose).

**Cells**. Vero E6 cells were obtained from European Collection of Authenticated Cell Cultures and maintained in minimal essential medium (MEM) (Gibco) supplemented with 10% heat-inactivated fetal bovine serum (FBS), 100 U of penicillin, and 100 μg of streptomycin and L-glutamine (PSG) (Gibco), 1X non-essential amino acid mix (NEAA) (Gibco), 25 mM HEPES in 5% $CO_2$ incubator. Calu-3 cells (Human lung epithelial cells derived from adenocarcinoma: ATCC-HTB-55) were maintained in Dulbecco's minimal essential medium (DMEM) (HiMedia) supplemented with 10% heat-inactivated fetal bovine serum (FBS), 100 U of penicillin and 100 μg of streptomycin and L-glutamine (PSG) (Gibco), 1X non-essential amino acid (NEAA) (Gibco), Expi293F, a derivative of human embryonic kidney (HEK293) cells, were maintained in suspension culture using serum-free Expi293 expression medium (Gibco).

**Viruses**. SARS-CoV-2 B.6 lineage and the Delta variant virus isolation has been described earlier[9,13,29]. SARS-CoV-2 Omicron isolate (sub-lineage BA.1) was obtained from Leo Poon[30]. SARS-CoV-2 variants were propagated in Vero E6 cells or Calu-3 cells[18] and virus passaging was limited to four passages. All virus stocks used in this study were verified by whole-genome sequencing using total RNA sample of the culture on Nanopore sequencing platform as described previously to confirm the variant[31].

**Quantitative nucleoprotein ELISA**. The *E. coli* Rosetta (DE3) bacterial host was used for expression of recombinant proteins. pET-28a(+) plasmid containing the codon-optimized nucleocapsid (N) gene from severe acute respiratory syndrome-related coronavirus 2 (SARS-CoV-2), Wuhan-Hu-1 (GenBank: MN908947) having N-terminal hexa-histidine affinity purification tag (BEI resources NR-53507) was used for expression of recombinant N protein. His-tagged N protein was purified by Ni-NTA chromatography as per previous report[32]. 96-well MaxiSorp ELISA plates (Nunc, Cat No. – 442404) were coated with 50 μl of 1 μg/mL purified N protein diluted in 1X PBS pH 7.4 and the plates were incubated for 1 h at room temperature (RT). Serum samples were inactivated by adding 10 μl of 10% Triton X-100 in 90 μl serum to obtain a final concentration of 1%. The plates were washed three times with 1X PBST (phosphate-buffered saline with 0.1% Tween 20) and non-specific binding sites were blocked by adding 200 μl of blocking buffer (5% non-fat dry milk powder (Bio-Rad, Cat No. – 1706404) prepared in 1X PBST) in each well and incubated at RT (23 ± 2 °C) for 1 h followed by one-time wash with 1X PBST. The serum samples were diluted 1:100 and 1:1000 in blocking buffer and 100 μl/well of each sample in two replicates was added to the antigen-coated plate. After 30 min at room temperature, the plate was washed six times using 1X PBST. After washing, 50 μl/well HRP-conjugated anti-human IgG (Jackson ImmunoResearch, cat no: 109-035-170) diluted 1:10,000 in blocking buffer was added and incubated at room temperature for 30 min followed by six washes with 1X PBST. 100 μl/well of 3,3′, 5,5′ tetramethylbenzidine substrate (BD Biosciences, Cat No. – 555214) was added for 10 min and the reaction was terminated using 1 M $H_2SO_4$ as stop solution. The intensity of the color was quantified by measuring absorbance in a microplate reader at 450 nm with 630 nm as reference wavelength. Antibody concentrations were calculated for each sample dilution by interpolation of the OD values on the 4-parameter logistic (4-PL) standard curve from in-house reference control (calibrated as a secondary standard using First WHO International reference standard reagent (20/136)) and adjusted according to their corresponding dilution factor using Gen5 software. The assay has a limit of quantitation of 15 binding antibody units/mL (BAU/mL) which was set after validation using pre-pandemic samples. All the samples with values below 15 BAU/mL were assigned a value of 7.5 BAU/mL for analysis. This assay has been validated in-house and accredited under ISO 17025:2017 standard.

**Quantitative RBD-ELISA**. The mammalian expression vector pcDNA™3.1 (+) containing the codon-optimized gene sequence of the receptor-binding domain (RBD, amino acids 328–531) of spike (S) glycoprotein from severe acute respiratory syndrome-related coronavirus 2 (SARS-CoV-2), Wuhan-Hu-1 (GenBank: MN908947) having an N-terminal mu-phosphatase signal sequence and C-terminal octa-histidine tag was obtained from BEI resources (NR-52422). RBD protein was expressed and purified in its fully glycosylated form through mammalian expression system (Expi293F cells) as per the previous reports[9,33]. 2 μg/mL Recombinant RBD antigen of SARS-CoV-2 spike protein in 1x PBS was coated onto 96-well MaxiSorp ELISA plates (50 μL/well) and incubated at 4 °C for 18–22 h. Antigen-coated plates were washed with wash buffer (1X PBST) and incubated by adding 200 μl of blocking buffer (3% non-fat dry milk powder in PBST) and incubated at RT (23 ± 2 °C) for 1 h. Serum samples inactivated with

Triton X-100 as described in the previous section and were diluted 1:50 or 1:500 in blocking buffer and 100 µl of diluted serum was added to each well in two replicates and incubated at RT (23 ± 2 °C) for 30–40 min followed by washes and secondary antibody incubation as described in the previous section. Substrate was added and OD was recorded, and data were analyzed as described in the previous section. The assay has a limit of quantitation of 12 binding antibody units/mL (BAU/mL) which was set after validation using pre-pandemic samples. All the samples with values below 12 BAU/mL were assigned a value of 6 BAU/mL for analysis. This assay has been validated in-house and accredited under ISO 17025:2017 standard.

**Virus microneutralization assay**. Virus neutralization was assessed by focus reduction neutralization titer assay using indicating virus isolates as described earlier with minor modifications for the Omicron variant staining[10,18]. Briefly, serum samples (two technical replicates each) were serially diluted from 1:20 to 1:640 in growth medium supplemented with 1% FBS, PSG and NEAA and virus neutralization was tested in Vero E6 cells. In the case of samples with RBD-ELISA-positive result, samples were pre-diluted to obtain values within the upper limit (1:640 dilution) of the assay and the final FRNT$_{50}$ value was calculated by considering the pre-dilution factor. Cells were incubated for 24 h for ancestral (B.6) and Delta (B.1.617.2) variants and for 32 h for the Omicron (B.1.1.529) variant. After incubation, cells were fixed with for-maldehyde solution and then stained with anti-spike RBD rabbit polyclonal antibody at 1:2000 dilution (Sino Biologicals, Cat. No. 40592-T62) for 1 h, followed by HRP-conjugated anti-rabbit antibody at 1:4000 dilution (Invitrogen, Cat. No. G-21234) for 1 h. For Omicron isolate, incubation was extended to 32 h and a 1:1000 dilution of anti-nucleocapsid primary antibody (GenScript, Cat. No. A02048-1) and 1:500 dilution of HRP-conjugated goat anti-mouse IgG secondary antibody (Invitrogen, Cat. No. A16072) was used for staining. Cells were washed with PBS and incubated with TrueBlue substrate (KPL Inc, USA, Cat. No. 5510-0030) for 10 min and washed with sterile MilliQ water. Microplaques developed after staining were quantified by AID iSpot reader (AID GmbH, Strassberg, Germany) using AID EliSpot 8.0 iSpot software. The raw data generated from the AID iSpot analyser in a 96-well format is pasted in a pre-defined protocol template for calculation of FRNT$_{50}$ by using log$_{10}$ transformed dilution value and neutralization percentages in an XY format. The Point-to-Point curve fit using a linear equation to fit each pair of data points was used to calculate the FRNT$_{50}$ value. 50% neutralization values were calculated using SoftMax Pro GxP software v7.7.1 (Molecular Devices). For the final analyses, samples below the lower limit of quantitation (LLOQ) (FRNT$_{50}$ of 20) were assigned a value of 10. Samples that had values over the upper limit of quantitation (ULOQ) even after pre-dilution were assigned the final ULOQ value obtained from the assay.

**Statistical analysis**. Data were analyzed using Microsoft Excel (v 16.16.27) and final graphs were prepared using GraphPad Prism (Version 9.3.1) software. Statistical significance was estimated by two-tailed, non-parametric tests as indicated in the respective figure legends.

**Reporting summary**. Further information on research design is available in the Nature Research Reporting Summary linked to this article.

## Data availability

All the data are presented in this manuscript. The virus strains used in this study have the following accession numbers: SARS-CoV-2—B.6 lineage (Genbank accession: MZ356566.1), SARS-CoV-2—B.1.1.617.2 (Genbank accession: MZ356566.1), and SARS-CoV-2—B.1.1.529 (BA.1 sub-lineage) (GISAID-accession: EPI_ISL_6716890 [https://www.epicov.org/epi3/frontend#31f622]). Source data are provided with this paper.

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

## Acknowledgements

We thank all the members of the bioassay lab for technical support. We thank Neha Garg and Shamsher Singh for data management. We thank all the participants who consented to enroll into the study. This work was supported by the Department of Biotechnology (DBT) through IndCEPI Mission (BT/MB/CEPI/2016), Translational Research Program (BT/PR30159/MED/15/188/2018). We acknowledge funding support from Global Immunology and Immune Sequencing for Epidemic Response, Bill and Melinda Gates Foundation (INV-030592). The funders had no role in study design, data collection and interpretation or the decision to submit the work for publication.

## Author contributions

S.D. and A.K.P. coordinated the study at the clinical site, generated all the clinical site data and contributed reagents. J.S., H.S., B.S., A.A., and P.S. performed experiments and analyzed the data. R.P. sequenced the virus isolates and analyzed the data. R.L. contributed resources, provided critical inputs in experimental design, and data analysis. G.R.M. conceived the study, designed the experiments, analyzed the data, and wrote the paper. All authors have reviewed and approved the final version of the manuscript.

## Competing interests

The authors declare no competing interests.
