## [Peer Review File · Nature Communications]

REVIEWER COMMENTS

Reviewer #1 (Remarks to the Author):

Das et al. analysed neutralising antibody responses against SARS-CoV-2 WT and Delta variant in pre-defined RBD-ELISA positive and negative individuals before vaccination and following one dose of BBV152 (inactivated COVID-19 vaccine). The authors found that neutralising responses against the WT strain and Delta variant were strongly enhanced in their pre-defined seronegative cohort following the single vaccine dose, while neutralising responses were modestly boosted in their pre-defined seropositive cohort. In addition, the authors showed that one dose of vaccine improved neutralising responses against Omicron.

It is relevant to analyse the enhancement of antibody responses and improvement of neutralising responses against variants of concerns following vaccination in baseline seronegative versus seropositive individuals. Several research groups have already analysed these vaccine-induced neutralising responses but usually it was after mRNA vaccination or viral vector-based vaccination. These data with BBV152 confirm the findings previously observed with the other vaccine platforms. The main weakness of this study is that previous infection with another variant either in individuals who became seronegative at the time of sample collection or in seropositive people cannot be excluded. It may impact on vaccine-induced neutralising responses. The methods used in this study look good to me.

Please find below a few comments to improve the paper.

Major comments:

- 1) To analyse the trajectory of antibody responses, it would be great to analyse antibody responses following the second dose as well. If the authors have some data following the second dose, it would be relevant to include them in order to have a “full story”.
- 2) Sometimes the paper may be confused with RBD-ELISA positive/negative individuals and seropositive/seronegative without any additional information in the text. Sometimes it is not fully clear if the authors always speak about the status at baseline. Can the authors make sure to be consistent in the text ?

3) As the authors used RBD ELISA to determine seropositive and seronegative individuals at baseline, could they show the cut-off for the RBD ELISA in panel A ? In addition, can they explain how they defined it in Methods ?

4) Can the authors also show the cut-off in panel B for the N ELISA and explain how they defined it in Methods ?

5) Line 68: the authors say that the GMT of RBD antibodies increased post-vaccination in baseline seropositive subjects. What is the GMT of RBD antibodies after vaccination in baseline seronegative individuals ?

6) Line 108: the authors wrote “Whether the modest boosting observed for omicron by BBV152 vaccination in seropositive individuals is protective or not warrants further studies from the ongoing booster vaccination campaign across the country.” This sentence is not clear based on panel G. In this sentence, do the authors mean RBD-ELISA positive individuals at baseline or individuals with neutralising responses to Omicron following vaccination ? In figure 1G, a number of individuals have neutralising responses against Omicron below the threshold before the single vaccine dose so they are seronegative at baseline. This sentence is confusing. Can the authors clarify ?

Minor comments:

7) For the N ELISA, it looks the authors did not use any blocking buffer. Is that correct ? Could they add to the text in Methods which diluent they used to dilute the samples and secondary anti-human IgG antibodies. In addition, could they add the reference of secondary antibodies ?

8) In Methods, the authors did not mention the secondary antibodies they used in RBD ELISA. Could they add this piece of information ?

9) In Figure 1 panels C and D, could the authors add a title “RBD-ELISA positive samples” and “RBD-ELISA negative samples” or “baseline seropositive” and “baseline seronegative” to the graphs ? In addition, could they define I and II in the legend ?

10) Panels F and G: the titles “First” and “Second” are not very clear as the authors analysed baseline samples and post one dose samples. Could they change this and be consistent in the figure ?

Reviewer #2 (Remarks to the Author):

The manuscript by Das, et al. describes the levels of neutralizing antibodies to the SARS-CoV-2 omicron variant following vaccination with a single dose of BBV152. The study is well designed and uses RBD and N ELISAs to confirm immunity and assess prior infection prior to vaccination. The authors show that natural infection of vaccination with BBV152 leads to increased neutralization of WT SARS-CoV-2 neutralization but a marginal and insignificant increase in neutralization of B.1.617.1. However, in individuals that were not previous exposed (or who were negative by RBD ELISA), there was an increase in neutralization of both WT and B.1.617.1 virus following a single dose of the vaccine. Finally, the authors detected a modest boosting of neutralization of the Omicron variant following vaccination. Notably this was a small difference, much smaller than the effects on B.1.617.1 neutralization. While many of the studies performed here have been done looking at other vaccines, this manuscript is unique in focusing on the BBV152 inactivated vaccine. Overall the paper is well written and the conclusions are justified by the data. However, I remain unconvinced of the significance of this work to the broader vaccine science community.

Minor

1. The method for indicating baseline vs post vaccine (I vs II) should be more clearly detailing in the figure legend.
2. Readability of manuscript would be improved though brief copy editing (mostly missing commas and articles)
3. It is unclear in some of the assays (between the methods and figure legend) how many replicates of each sample were performed.
4. The authors refer to a non-peer reviewed pre-print utilizing the neutralization assay methods, which itself cite a protocol previously performed in another manuscript. Neither protocol (the one provided in the current manuscript or the preprint) indicates whether the virus and serum were preincubated and if so, under what conditions. The manuscript being reviewed should add Bewley, et al. to the citation on line 83 to make it more clear to readers where to find details necessary for replicating the data.

Reviewer #3 (Remarks to the Author):

SUMMARY

In this study, Das et al examine SARS-CoV-2 RBD and nuclear capsid antibody responses before and after vaccination with a single dose of the BB V152 Covid-19 vaccine, based on inactivated SARS-CoV-2, in a group of 94 adults drawn from a population with relatively high prior exposure to Covid. They performed both binding and neutralizing antibody studies to the delta and omicron variants in recent circulation.

This study addresses an important topic and provides an interesting review of antibody responses following vaccination with BBV152 and the potentially important observation that those with relatively high binding antibody responses at baseline appear to have minimal increase in some cases a decline in antibody responses 4 weeks following vaccination, raising the concern of anergy, at least in those individuals who had evidence of prior infection.

However, the results section of this manuscript is on systematic and a bit confusing, and as a result is unnecessarily difficult to follow. This section must be carefully revised for clarity, and a logical presentation of the results by each variable being examined.

In addition, one of the fundamental comparisons presented was neutralizing antibody responses in those with and without baseline RBD specific-antibodies. This points towards a comparison of SARS-CoV-2-naïve versus SARS-CoV-2 exposed participants and how they are antibody responses defer following vaccination which is important to examine. However the proper comparison for this would be those with any Covid antibodies (RBD or nucleocapsid) versus those with no Covid antibodies. The authors have these data and a direct comparison of these 2 groups would be helpful.

The major and minor comments must be addressed before any consideration of publication should be given.

MAJOR COMMENTS

1. In the methods section, the authors do not provide sufficient information on how participants were selected for study, nor on what population was used for this research. Some information is provided under human samples in the supplement but individuals presenting for vaccination at the ESIC might be students, employees, or vaccination might be open to the public at this location. Some understanding of what population was drawn from 2 perform the study would help the reader in understanding any potential biases and should be provided.

2. In the population under study, line 58, it would appear, but is not explicitly stated that individuals were sampled prior to receipt of any Covid-19 vaccine. If correct, this should be made explicit. If this is not the case, it should be clarified that some individuals may have been vaccinated prior to enrollment.

3. In the method section, line 64, the authors mention that 94 samples were positive for in antibodies, suggesting exposure in the second wave. In this setting, antibodies for nucleocapsid are consistent with either infection or receipt of BBV 152 (Covid vaccine based on inactivated virus containing nuclear capsid). However to properly interpret this the authors must let us know if these individuals could have previously received a vaccine or if this was the first vaccination.

4. The section dealing with neutralizing antibody assay results between lines 71 and 94 is somewhat disorganized and difficult to follow, as it is presented largely as a collection of individual observations rather than a systematic walk-through of the data by theme. For example, in line 77 the authors start by saying "the FRNT data were further analyzed based on the seropositivity status at baseline", setting up an expectation of a step-by-step comparison between the results seen in those who did in those who did not have detectable antibodies prior to vaccination. However, what follows is a bit of a jumble of individual observations sometimes mentioning baseline status, sometimes mentioning antibodies against wild-type and delta, sometimes mentioning a count of changes out of a denominator in 1 subgroup but no parallel analysis in the opposite subgroup, etc. This section should be carefully reviewed and reorganized to systematically present the data in a way that can be easily followed, and interpretation of the data should be reserved for the end of the paragraph or the discussion section.

5. An analysis of the ratio of neutralizing antibody titer to total SARS-CoV-2 specific (or RBD specific) antibody quantity would be helpful to assess changes in antibody quality prior to and after vaccination.

6. In figure 1C and figure 1 the authors compare neutralizing antibody responses (FRNT 50) between those with and without RBD-specific antibodies at baseline. Presumably the latter group has not had exposure to SARS-CoV-2. However, the appropriate group would appear to be those without either RVD-specific or nucleocapsid-specific antibodies at baseline. The authors have these data and this should have been the comparison group since it would more reliably distinguish those with and without prior Covid exposure. In this regard, it would be helpful to see a 2x2 analysis of how many individuals were positive for RBD-specific antibodies, nucleocapsid-specific antibodies, neither or both.

7. Age is a significant factor in the magnitude of the antibody response. Were there any age differences between naïve and exposed groups prior to vaccination?

MINOR COMMENTS

8. In figure 1, thus panel a referral to binding antibody responses to wild-type virus? This should be stated

9. Could the authors please clarify what the differences between what is shown in figure 1C and 1F

10. Introduction, Line 41: The authors used the phrase "after the second wave". This should be clarified to indicate second wave of infection with SARS-CoV-2 (as opposed to any additional rounds of vaccination, etc.) and give an approximate timeframe for this event.

11. In a few cases, an article is missing before the noun. This occurs on lines 44 (before "majority"), 50 (before "routine vaccination program"), 52 (before "majority") etc.

12. Line 209 "was" should be "were".

13. Supplement line 36 "was" should be "were".

14. The authors should review the manuscript closely for any grammatical issues.

We thank the reviewers for their valuable comments to help improve our manuscript. We have modified the manuscript as per the comments and providing point-wise responses to the comments.

Reviewer #1 (Remarks to the Author):

Das et al. analysed neutralising antibody responses against SARS-CoV-2 WT and Delta variant in pre-defined RBD-ELISA positive and negative individuals before vaccination and following one dose of BBV152 (inactivated COVID-19 vaccine). The authors found that neutralising responses against the WT strain and Delta variant were strongly enhanced in their pre-defined seronegative cohort following the single vaccine dose, while neutralising responses were modestly boosted in their pre-defined seropositive cohort. In addition, the authors showed that one dose of vaccine improved neutralising responses against Omicron.

It is relevant to analyse the enhancement of antibody responses and improvement of neutralising responses against variants of concerns following vaccination in baseline seronegative versus seropositive individuals. Several research groups have already analysed these vaccine-induced neutralising responses but usually it was after mRNA vaccination or viral vector-based vaccination. These data with BBV152 confirm the findings previously observed with the other vaccine platforms. The main weakness of this study is that previous infection with another variant either in individuals who became seronegative at the time of sample collection or in seropositive people cannot be excluded. It may impact on vaccine-induced neutralising responses. The methods used in this study look good to me.

Response: We agree that our study does not rule out the possibility of past exposure in subjects who were seronegative by RBD ELISA. Please note that, 22 out of 23 RBD-ELISA negative samples had neutralizing antibodies for the Delta variant in FRNT assay. However, the levels of these antibodies were about seven- to eight-fold lower than that in the ELISA-positive samples (Table 1) suggesting that the antibody levels were below the level of detection of ELISA. Half-life of antibodies, as reported by our and other studies is at least six months after natural infection or vaccination (<https://doi.org/10.1101/2022.01.28.22269990>; <https://doi.org/10.1101/2022.02.23.22271381>). Therefore, absence of detectable levels of antibodies by ELISA but presence of neutralizing antibodies would mean past exposure any time in the six months prior to sample collection or these are neutralizing antibodies that bind outside the RBD region. We have discussed this aspect in both the results and discussion section. Please see lines 101-106, 157-169.

Prior to Delta/Kappa variant which caused the second wave in India, the dominant circulating variant of concern was the Alpha variant. To address reviewer's concern, we randomly selected 17 each of RBD-positive and RBD-negative samples to test for neutralizing antibodies against the Alpha variant by FRNT assay. The geometric mean titers (GMT) of neutralizing antibodies for Alpha and Delta variants are shown below:

FRNT₅₀ titers for the B.1.1.7 (Alpha) and B.1.617.2 (Delta) variant in baseline samples that were either positive or negative in RBD ELISA (n=17 each). LLOQ: Lower limit of quantitation of the assay. Geometric mean value of FRNT₅₀ titers of each group is indicated at the top.

As has been reported by many studies now, natural infection or vaccination would result in induction of antibodies that offer cross-protection for other variants but the level of antibody titers may vary from variant to variant. We observed higher FRNT values for the Delta variant as compared to the Alpha variant (above figure) indicating a likely exposure with the Delta variant in the second wave prior to vaccination. Also note that the overall seroprevalence in India till Feb 2021 was 20.7% which increased to 69.2% by August 2021 (<https://doi.org/10.1016/j.ijid.2021.12.353>) despite the low vaccination coverage (11% completely vaccinated, 25% with one dose) by the end of August 2021.

Please find below a few comments to improve the paper.

Major comments:

1) To analyse the trajectory of antibody responses, it would be great to analyse antibody responses following the second dose as well. If the authors have some data following the second dose, it would be relevant to include them in order to have a “full story”.

Response: We agree that antibody response after second dose would have made the story more complete. Our research question was focussed on understanding how prior exposure to SARS-CoV-2 affects response to vaccination with BBV152 which mimics the booster dose scenario. The study relied on collecting samples from general subjects who were visiting the vaccination centre for routine vaccination. Therefore, it was possible to enrol participants during their visit for first and second dose of vaccine. Collecting samples after second dose would have required us to actively follow up subjects and arrange for their visit to the study site for sample collection or visit homes to collect sample, both of which was difficult to manage logistically.

2) Sometimes the paper may be confused with RBD-ELISA positive/negative individuals and seropositive/seronegative without any additional information in the text. Sometimes it is not fully clear if the authors always speak about the status at baseline. Can the authors make sure to be consistent in the text ?

Response: We have now addressed this issue by maintaining consistency in the figures and text by using “baseline” for the first sample prior to vaccination and “one dose” for the sample collected about 4 weeks later (just before receiving the second dose of BBV152). We have further separated ELISA and FRNT data into two figures for clarity and show ELISA titers for RBD and N separately based on positivity at baseline for each. Please see the modified figure 1 and 2 and the results section (line 62-85).

3) As the authors used RBD ELISA to determine seropositive and seronegative individuals at baseline, could they show the cut-off for the RBD ELISA in panel A ? In addition, can they explain how they defined it in Methods ?

Response: We have now shown the lower limit of quantitation (LLOQ) in all the figures by a dotted line. We have revised the methods section to clearly describe the process. The cut-off was determined by testing pre-pandemic samples and assay was calibrated and validated using the first WHO reference reagent. Please see lines 248-254.

4) Can the authors also show the cut-off in panel B for the N ELISA and explain how they defined it in Methods ?

Response: This has been addressed. Please see our response above.

5) Line 68: the authors say that the GMT of RBD antibodies increased post-vaccination in baseline seropositive subjects. What is the GMT of RBD antibodies after vaccination in baseline seronegative individuals ?

Response: We have now included data as new Figure 1b and described in the results section. The GMT of antibodies was 19.4 (95% CI: 12, 32) after one dose in samples that were RBD-negative at baseline. Please see lines 74-77.

6) Line 108: the authors wrote “Whether the modest boosting observed for omicron by BBV152 vaccination in seropositive individuals is protective or not warrants further studies from the ongoing booster vaccination campaign across the country.” This sentence is not clear based on panel G. In this sentence, do the authors mean RBD-ELISA positive individuals at baseline or individuals with neutralising responses to Omicron following vaccination ? In figure 1G, a number of individuals have neutralising responses against Omicron below the threshold before the single vaccine dose so they are seronegative at baseline. This sentence is confusing. Can the authors clarify ?

Response: Please note that we were expecting at least 15 to 20-fold reduction in neutralizing antibody titers for the Omicron variant based on reports published by our and other groups. Therefore, we randomly selected 55 out of the total 94 samples which had detectable levels of neutralizing antibodies for the delta variant after one dose of vaccination with the sole aim of detecting neutralizing antibodies for the Omicron variant. Therefore, the data were not segregated into RBD-ELISA positive or negative samples. We have clarified this further in the results section. Please see lines 133-136.

Minor comments:

7) For the N ELISA, it looks the authors did not use any blocking buffer. Is that correct ? Could they add to the text in Methods which diluent they used to dilute the samples and secondary anti-human IgG antibodies. In addition, could they add the reference of secondary antibodies ?

Response: We thank the reviewer for bringing this to our notice. Blocking buffer was used in both the ELISAs but the concentration of blocking agent was different. The methods section has been revised to indicate this. Please see lines 235-236 and 266-267.

8) In Methods, the authors did not mention the secondary antibodies they used in RBD ELISA. Could they add this piece of information ?

Response: We thank the reviewer for bringing this to our notice. We have now provided a complete table of all reagents used in the study as a Key Resource Table in the supplementary information.

9) In Figure 1 panels C and D, could the authors add a title “RBD-ELISA positive samples” and “RBD-ELISA negative samples” or “baseline seropositive” and “baseline seronegative” to the graphs ? In addition, could they define I and II in the legend ?

Response: The figures have been modified as per the advice. Please see the revised figure 1 and 2.

10) Panels F and G: the titles “First” and “Second” are not very clear as the authors analysed baseline samples and post one dose samples. Could they change this and be consistent in the figure ?

Response: We have now changed this to Figure 2d and 2e. The figures have been modified as per the advice. Please see the revised figure.

Reviewer #2 (Remarks to the Author):

The manuscript by Das, et al. describes the levels of neutralizing antibodies to the SARS-CoV-2 omicron variant following vaccination with a single dose of BBV152. The study is well designed and uses RBD and N ELISAs to confirm immunity and assess prior infection prior to vaccination. The authors show that natural infection of vaccination with BBV152 leads to increased neutralization of WT SARS-CoV-2 neutralization but a marginal and insignificant increase in neutralization of B.1.617.1. However, in individuals that were not previous exposed (or who were negative by RBD ELISA), there was an increase in neutralization of both WT and B.1.617.1 virus following a single dose of the vaccine. Finally, the authors detected a modest boosting of neutralization of the Omicron variant following vaccination. Notably this was a small difference, much smaller than the effects on B.1.617.1 neutralization. While many of the studies performed here have been done looking at other vaccines, this manuscript is unique in focusing on the BBV152 inactivated vaccine. Overall the paper is well written and the conclusions are justified by the data. However, I remain unconvinced of the significance of this work to the broader vaccine science community.

Response: Our research question was focussed on understanding how prior exposure to SARS-CoV-2 affects response to vaccination with BBV152. The sero-status of our study participants mimics the booster dose vaccination scenario. The second wave of COVID-19 infections between April 2021 to June 2021 led to overall increase in sero-positivity to >70% in India. Most of the Indian population has been vaccinated post-second wave and there have been no systematic, prospective studies to measure the effect of prior SARS-CoV-2 infection on vaccine response, particularly with the inactivated BBV152 vaccine which was one of the two vaccines approved for use in India in 2021. Furthermore, the current booster vaccination policy advocates homologous boosters and there is no data demonstrating the efficacy of booster doses in subjects with prior infection and whether these booster doses would generate antibodies against the currently circulating virus strains (BA.1 and BA.2 sub-lineages of Omicron). Therefore, our study provides information on the antibody responses upon

BBV152 vaccination in individuals who were naturally infected with SARS-CoV-2 prior to receiving their first dose of vaccine. Our results may influence change in policy to allow heterologous booster doses which have been shown to be more effective in subjects vaccinated with another inactivated vaccine (CoronaVac) in other countries.

Minor

1. The method for indicating baseline vs post vaccine (I vs II) should be more clearly detailing in the figure legend.

Response: We thank the reviewer for pointing this out. We have rectified this in the new Figure 2.

2. Readability of manuscript would be improved though brief copy editing (mostly missing commas and articles)

Response: We have carefully revised the manuscript and corrected typographical errors.

3. It is unclear in some of the assays (between the methods and figure legend) how many replicates of each sample were performed.

Response: Neutralization assays are performed with two replicates of each sample serially diluted from 1:20 to 1:640. ELISA was performed with two different pre-dilutions of each of the samples in two technical replicates each. The methods section has been revised to indicate this.

4. The authors refer to a non-peer reviewed pre-print utilizing the neutralization assay methods, which itself cite a protocol previously performed in another manuscript. Neither protocol (the one provided in the current manuscript or the preprint) indicates whether the virus and serum were preincubated and if so, under what conditions. The manuscript being reviewed should add Bewley, et al. to the citation on line 83 to make it more clear to readers where to find details necessary for replicating the data.

Response: We have added the reference of Bewley et.al as advised in both results and methods section.

Reviewer #3 (Remarks to the Author):

SUMMARY

In this study, Das et al examine SARS-CoV-2 RBD and nuclear capsid antibody responses before and after vaccination with a single dose of the BB V152 Covid-19 vaccine, based on inactivated SARS-CoV-2, in a group of 94 adults drawn from a population with relatively high prior exposure to Covid. They performed both binding and neutralizing antibody studies to the delta and omicron variants in recent circulation.

This study addresses an important topic and provides an interesting review of antibody responses following vaccination with BBV152 and the potentially important observation that those with relatively high binding antibody responses at baseline appear to have minimal increase in some cases a decline in antibody responses 4 weeks following vaccination, raising the concern of anergy, at least in those individuals who had evidence of prior infection.

However, the results section of this manuscript is on systematic and a bit confusing, and as a

result is unnecessarily difficult to follow. This section must be carefully revised for clarity, and a logical presentation of the results by each variable being examined.

In addition, one of the fundamental comparisons presented was neutralizing antibody responses in those with and without baseline RBD specific-antibodies. This points towards a comparison of SARS-CoV-2-naïve versus SARS-CoV-2 exposed participants and how they are antibody responses defer following vaccination which is important to examine. However the proper comparison for this would be those with any Covid antibodies (RBD or nucleocapsid) versus those with no Covid antibodies. The authors have these data and a direct comparison of these 2 groups would be helpful.

Response: We have now revised the result section by sequentially describing the ELISA data in RBD-positive and RBD-negative subjects followed by N-positive and N-negative subjects. This is now presented as new Figure 1. We have moved the neutralising antibody data to Figure 2 which shows the GMT values of neutralizing antibodies in RBD-positive and RBD-negative samples (Figure 2a and 2b).

The major and minor comments must be addressed before any consideration of publication should be given.

MAJOR COMMENTS

1. In the methods section, the authors do not provide sufficient information on how participants were selected for study, nor on what population was used for this research. Some information is provided under human samples in the supplement but individuals presenting for vaccination at the ESIC might be students, employees, or vaccination might be open to the public at this location. Some understanding of what population was drawn from 2 perform the study would help the reader in understanding any potential biases and should be provided.

Response: The enrolled subjects were general public who visited the vaccination centre for their routine vaccination as per the National COVID-19 immunization program. All participants received their first COVID-19 vaccine while enrolling into this study. Please see lines 62-64.

2. In the population under study, line 58, it would appear, but is not explicitly stated that individuals were sampled prior to receipt of any Covid-19 vaccine. If correct, this should be made explicit. If this is not the case, it should be clarified that some individuals may have been vaccinated prior to enrolment.

Response: We have now clarified this in the section in results: "Human Samples". None of the participants had received COVID-19 vaccine prior to enrolment. Please see lines 62-66.

3. In the method section, line 64, the authors mention that 94 samples were positive for in antibodies, suggesting exposure in the second wave. In this setting, antibodies for nucleocapsid are consistent with either infection or receipt of BBV 152 (Covid vaccine based on inactivated virus containing nuclear capsid). However to properly interpret this the authors must let us know if these individuals could have previously received a vaccine or if this was the first vaccination.

Response: All participants received their first dose of COVID-19 vaccine while enrolling into this study. The first sample was collected prior to receipt of the first dose of BBV 152

vaccine.

4. The section dealing with neutralizing antibody assay results between lines 71 and 94 is somewhat disorganized and difficult to follow, as it is presented largely as a collection of individual observations rather than a systematic walk-through of the data by theme. For example, in line 77 the authors start by saying "the FRNT data were further analyzed based on the seropositivity status at baseline", setting up an expectation of a step-by-step comparison between the results seen in those who did in those who did not have detectable antibodies prior to vaccination. However, what follows is a bit of a jumble of individual observations sometimes mentioning baseline status, sometimes mentioning antibodies against wild-type and delta, sometimes mentioning a count of changes out of a denominator in 1 subgroup but no parallel analysis in the opposite subgroup, etc. This section should be carefully reviewed and reorganized to systematically present the data in a way that can be easily followed, and interpretation of the data should be reserved for the end of the paragraph or the discussion section.

Response: We thank the reviewer for this comment. We have tried to address this concern in the revised result section. Table 1 provides all the comparisons within and between the groups and the numbers were not repeated in the text to avoid redundancy. The information provided in the revised results section first describes overall change in neutralization antibody titers for both B.6 and B.1.617.2 lineage viruses in 94 samples and refers to Table 1 for values and statistical tests. Then, we sequentially describe the change in antibody titers for B.6 and Delta variant in RBD-positive and RBD-negative samples after one dose of BBV152 vaccination relative to baseline samples. Please see the revised results section, lines 91-110.

5. An analysis of the ratio of neutralizing antibody titer to total SARS-CoV-2 specific (or RBD specific) antibody quantity would be helpful to assess changes in antibody quality prior to and after vaccination.

Response: We thank the reviewer for this comment. We calculated the ratio of FRNT₅₀ values of B.6 or the Delta variant to RBD-antibody titers. We have added the description of these analyses in the results section. Please see lines 115-126. A new supplementary Figure 2 has been added representing this analysis.

6. In figure 1C and figure 1 the authors compare neutralizing antibody responses (FRNT 50) between those with and without RBD-specific antibodies at baseline. Presumably the latter group has not had exposure to SARS-CoV-2. However, the appropriate group would appear to be those without either RVD-specific or nucleocapsid-specific antibodies at baseline. The authors have these data and this should have been the comparison group since it would more reliably distinguish those with and without prior Covid exposure. In this regard, it would be helpful to see a 2x2 analysis of how many individuals were positive for RBD-specific antibodies, nucleocapsid-specific antibodies, neither or both.

Response: Please see the 3X3 table below which provides clarity on the antibody status of 94 subjects. We have revised figure 1 and indicated the number of samples above each figure panel in Figure 1a-1d.

Table 1: Samples and serostatus

ELISA		Anti-nucleocapsid antibody		
		Positive	Negative	Equivocal
Anti-RBD antibody	Positive	53	7	7
	Negative	2	20	1
	Equivocal	0	4	0

Of the 23 samples that were negative at baseline in RBD-ELISA, only two samples were positive in N-ELISA and one was equivocal. We analysed the data by excluding these three samples and obtained almost similar data. Please see the figure below (n=20):

7. Age is a significant factor in the magnitude of the antibody response. Were there any age differences between naïve and exposed groups prior to vaccination?

Response: We found no difference in age between naïve and exposed groups prior to vaccination. Please see the figure here which shows the median age with range.

MINOR COMMENTS

8. In figure 1, thus panel a referral to binding antibody responses to wild-type virus? This should be stated

Response: We have revised the methods section as described here: *The mammalian expression vector pcDNA™3.1 (+) containing the codon-optimized gene sequence of the receptor-binding domain (RBD, amino acids 328 to 531) of spike (S) glycoprotein from severe acute respiratory syndrome-related coronavirus 2 (SARS-CoV-2), Wuhan-Hu-1 (GenBank: MN908947) having an N-terminal mu-phosphatase signal sequence and C-terminal octa-histidine tag was obtained from BEI resources (NR-52422).*

9. Could the authors please clarify what the differences between what is shown in figure 1C and 1F

Response: Figure 1F had a mix of participants randomly selected from Figure 1C and Figure 1D (new Figure 2a and 2b) for testing neutralizing antibodies for Omicron. New Figure 2d shows the neutralizing antibodies for Delta variant for these 55 samples. The results section has been revised to indicate this. Please see lines 133 to 136.

10. Introduction, Line 41: The authors used the phrase "after the second wave". This should be clarified to indicate second wave of infection with SARS-CoV-2 (as opposed to any additional rounds of vaccination, etc.) and give an approximate timeframe for this event.

Response: We have revised the text as suggested. Please see lines 42-43.

11. In a few cases, an article is missing before the noun. This occurs on lines 44 (before "majority"), 50 (before "routine vaccination program"), 52 (before "majority") etc.

12. Line 209 "was" should be "were".

13. Supplement line 36 "was" should be "were".

14. The authors should review the manuscript closely for any grammatical issues.

Response: We have carefully corrected typographical errors.

REVIEWERS' COMMENTS

Reviewer #1 (Remarks to the Author):

The authors perfectly replied to the comments. The clarity of the figures and the manuscript has been significantly improved. I do not have any additional comments.

Reviewer #3 (Remarks to the Author):

Das et al have made substantial changes to this manuscript in response to the reviews that have been offered. The manuscript has significantly improved as a result of these changes, and overall these results do provide helpful new information on immune responses in baseline seropositive and seronegative individuals receiving a single dose of a COVID-19 vaccine based on inactivated virus construct.

My one remaining criticism is with the phrase "Antibody boosting correlated negatively with the levels of pre-existing antibodies" which appears on line 169 and a discussion. In fact, Table1 shows that participants who were RBD-positive at baseline had a rise in mean FRNT50 against B.1.617.2 from 531.9 to 783.2 (absolute increase = 251.3, fold increase =1.5) while participants who were RBD-negative at baseline had a rise in mean FRNT50 from 63.9 to 146.6 (absolute increase = 82.7, fold increase = 2.3).

Therefore, while the fold increase was less for those who were RBD-positive at baseline, the absolute rise was greater in this group. If the absolute level of neutralizing antibody titers correlate with protection, the overall protective effect of a single dose of BBV152 would likely still be greater in the group that is seropositive at baseline. The authors should take this into account in their interpretation of antibody responses.

Reviewer #1 (Remarks to the Author):

The authors perfectly replied to the comments. The clarity of the figures and the manuscript has been significantly improved. I do not have any additional comments.

Response: We thank the reviewer for the comments.

Reviewer #3 (Remarks to the Author):

Das et al have made substantial changes to this manuscript in response to the reviews that have been offered. The manuscript has significantly improved as a result of these changes, and overall these results do provide helpful new information on immune responses in baseline seropositive and seronegative individuals receiving a single dose of a COVID-19 vaccine based on inactivated virus construct.

My one remaining criticism is with the phrase "Antibody boosting correlated negatively with the levels of pre-existing antibodies" which appears on line 169 and a discussion. In fact, Table1 shows that participants who were RBD-positive at baseline had a rise in mean FRNT50 against B.1.617.2 from 531.9 to 783.2 (absolute increase = 251.3, fold increase =1.5) while participants who were RBD-negative at baseline had a rise in mean FRNT50 from 63.9 to 146.6 (absolute increase = 82.7, fold increase = 2.3).

Therefore, while the fold increase was less for those who were RBD-positive at baseline, the absolute rise was greater in this group. If the absolute level of neutralizing antibody titers correlate with protection, the overall protective effect of a single dose of BBV152 would likely still be greater in the group that is seropositive at baseline. The authors should take this into account in their interpretation of antibody responses.

Response: We thank the reviewer for the comments. We agree that despite the smaller increase in antibody levels in RBD-positive individuals, the absolute levels of antibodies is likely to be sufficient for protection. We have now revised the text in results section in lines 123-125 to include a statement as per the reviewer's suggestion. From our results and based on other reports published recently, we speculate that after the second dose of the vaccine, the antibody levels in the RBD-negative group may be boosted further. However, in individuals with high levels of baseline antibodies from prior infection, a second dose of BBV152 vaccine may not have any impact on the antibody responses. Whether this would be specific to inactivated vaccines or is it due to exhaustion of the immune system remains to be investigated.